# The efficacy and safety of lignocaine-embedded dissolvable microneedle versus EMLA for topical analgesia in adults undergoing venepuncture: A single-centre, parallel-group, double-blind randomised clinical trial protocol in a tertiary care setting

**Muhammad Irfan Abdul Jalal**[1]*, **Mun Yin Yen**[2], **Lam Chenshen**[2],
**Mae-Lynn Catherine Bastion**[2], **Chua Xin Yun**[3], **Sharipah Salwa Abdul Razak**[4],
**Sharmilah Kuppusami**[5], **Arifah Syahirah Abdul Rahman**[6], **Mohd Eusoff Azizol Nashriby**[6],
**Nurul Ashikin A. Rahim**[6], **Poh Choon Ooi**[6], **Muhamad Ramdzan Buyong**[6],
**Mohd Ambri Mohamed**[6], **Chang Fu Dee**[6], **Azlan Azrul Hamzah**[6], **Fook-Choe Cheah**[7]*

**1** UKM Medical Molecular Biology Institute (UMBI), Hospital Canselor Tuanku Muhriz (HCTM), Universiti Kebangsaan Malaysia (UKM), Kuala Lumpur, Malaysia, **2** Department of Ophthalmology, Hospital Canselor Tuanku Muhriz (HCTM), Universiti Kebangsaan Malaysia (UKM), Kuala Lumpur, Malaysia, **3** Department of Pharmacy, Hospital Canselor Tuanku Muhriz (HCTM), Universiti Kebangsaan Malaysia (UKM), Kuala Lumpur, Malaysia, **4** Department of Diagnostic Laboratory Services, Hospital Canselor Tuanku Muhriz (HCTM), Universiti Kebangsaan Malaysia (UKM), Kuala Lumpur, Malaysia, **5** Toxicology Division, Forensic Science Analysis Centre, Department of Chemistry Malaysia (KIMIA Malaysia), Selangor, Malaysia, **6** Institute of Microengineering and Nanoelectronics (IMEN), Level 4, Research Complex, Universiti Kebangsaan Malaysia (UKM), Selangor, Malaysia, **7** Department of Paediatrics, Faculty of Medicine, Hospital Canselor Tuanku Muhriz (HCTM), Universiti Kebangsaan Malaysia (UKM), Kuala Lumpur, Malaysia

\* irfan.abduljalal@ukm.edu.my (MIAJ); cheahfc@hctm.ukm.edu.my (FCC)

## Abstract

Venepuncture-associated pain is a major source of distress commonly experienced by adult patients in day-to-day clinical practice. Topical lignocaine application before venepuncture may address this issue, but this delivery approach may be suboptimal. Hence, we aim to investigate the safety and efficacy of a novel lignocaine-embedded transdermal microneedle array patch (LEMAP) in facilitating transcutaneous lignocaine delivery to reduce procedural-related pain in adults undergoing venepuncture in a tertiary-care outpatient clinic setting. This is an investigator-initiated, single-centre, active-controlled, double-blind, randomised superiority trial divided into two distinct stages. Twenty (single-group LEMAP recipients) and 144 adult patients (72 per group; randomised to either LEMAP (intervention) or 5% EMLA patch (control) applied on antecubital fossa, near the venepuncture site, for 30 minutes) aged 18 years and above requiring routine venepuncture will be recruited from the Ophthalmology Outpatient Clinic, Hospital Canselor Tuanku Muhriz, for stage I and II of this trial, respectively. For the stage I trial, the safety endpoints are lignocaine's pharmacokinetic parameters and clinical adverse events. In the stage II trial, the primary endpoints are the

**Data availability statement:** Deidentified research data will be made publicly available when the study is completed and published.

**Funding:** This study is supported by the Technology Development Fund 2 (TED2) Grant (Grant number: MOSTI.D (S) 600-4/19/15) funded by the Ministry of Science, Technology and Innovation (MOSTI), Malaysia and UKM Faculty of Medicine Grant (Grant number: FF-2021-401). The funders had no role in study design, data collection and analysis, decision to publish, or preparation of the manuscript.

**Competing interests:** I have read the journal's policy and the authors of this manuscript have the following competing interests: AAH, DCF, MAM, MRB and PCO are named inventors on a patent application for LEMAP (UKM.IKB.800-4/1/5849), owned by UKM. There is no licence or option agreement in place. No author has any personal financial interest, royalties, equity, or paid role related to this technology. The remaining authors declare no competing interests.

**Abbreviations:** LEMAP, Lignocaine-Embedded Microneedle Array Patch; VAS, Visual Analogue Scale; SCAI, Skin Conductance Algesimeter Index.

venepuncture-associated pain experience, which will be evaluated using the Visual Analogue Scale (VAS) and the skin conductance algesimeter index (SCAI) scores at one-minute post-venepuncture. Non-linear mixed-effect model and multiple linear regression will be used to analyse the stage I and II trial outcomes, respectively. The trial protocol has been registered with the clinicaltrial.gov registry (ID: NCT05694858) and adheres to the SPIRIT 2025 reporting guideline. All trial participants will provide written informed consent, which the trial investigators will obtain before trial enrollment and randomisation. The trial findings will be disseminated via peer-reviewed publications and presentations at international conferences and shared with participants via the web-based trial notification system..

## Introduction

Venepuncture is one of the most commonly encountered invasive medical procedures that causes significant traumatic pain and stress in a clinical treatment facility. The anxiety and apprehension toward hypodermic needles may be exacerbated in patients with chronic conditions who require frequent venepuncture. This, in the long term, may affect their emotional well-being, which could lead to lower patient compliance and treatment acceptability [1,2].

With the advent of microfabrication technology, a revolutionary solution in effective transdermal drug delivery has been introduced with the creation of a microneedle (MN) patch. MN is a medical device that is composed of tiny micron-sized needles aligned in out-of-plane protruded arrays that can be impregnated with bioactive drug substances [3]. These micron-sized needles create multiple transdermal microchannels when they puncture through the skin stratum corneum. Consequently, this innovative means of drug delivery enhances transcutaneous drug absorption, resulting in a faster onset of therapeutic action [4].

To date, the studies investigating the effects of dissolving microneedles on delivering topical anaesthetics on adult patients at venepuncture are scarce. The closest studies include evaluating the efficacy of non-drug-loaded microneedles for alleviating pain in adult patients undergoing routine peripheral vein-puncturing procedures. Rzhevskiy et al. demonstrated in a clinical trial investigating the efficacy of lignocaine delivered via a hollow microneedle (MicronJet600) prior to peripheral venous cannulation. They showed a significant 11-fold VAS score reduction in adults undergoing routine peripheral venous cannulation when 2% lignocaine was intradermally administered using MicronJet600 compared to no anaesthetic pretreatment (mean VAS score: 3.6 (MJ + L combination) vs 39.7 (placebo); Cohen's d: −.43 (95% CI −48, −3.9)) [5]. In addition, Ornelas and colleagues conducted a randomised, single-blinded, parallel-group clinical trial to evaluate the effect of microneedle pre-treatment in hastening the onset of cutaneously-applied 4% lignocaine cream. They demonstrated that the microneedle-assisted 4% lignocaine delivery had shortened the application time from 60 to 30 minutes and significantly minimised the pain induced by needle lancet at 30 mins compared to sham patch [VAS score (mean SD): microneedle: 4 1.3 mm; sham: 14.4 3.8 mm] [6]. In contrast, Gupta et al. compared

the efficacy of lignocaine injections administered by hollow borosilicate-glass microneedle with the conventional hypodermic needle in 15 healthy adults. They established that both lignocaine delivery systems produce similar local anaesthetic effects (p > 0.05) across all time points, but the hollow microneedle recipients reported significantly better dermal analgesic effects compared to the recipients of conventional hypodermic needles [7].

Based on these findings, the utility of microneedles for effective anaesthetic delivery and pain amelioration appears promising. However, the validity and generalisability of the aforementioned evidence are challenged by several methodological shortcomings. For instance, the efficacy of transdermal microneedle in alleviating venepuncture-related pain has not been investigated in a clinical trial setting. Besides, Ornelas et al's RCT only included male participants, a significant methodological weakness that hampers the results' generalisability to female patients [6]. Apart from that, Gupta et al. indicated that the studied hollow borosilicate-glass microneedle prototype suffers from undesirable lignocaine leakage when it was applied to their subjects [7].

Our recently concluded randomised cross-over clinical trial has demonstrated the superiority of a stand-alone microneedle array patch design over sham patch in facilitating 5% EMLA cream delivery in transfusion-dependent thalassemia children requiring regular intravenous cannulation [8,9]. However, based on the trial's findings, we also discovered that the microneedle array patch design could still be further optimised by incorporating the local anaesthetic agent directly into the microneedle matrix, which will result in a more optimal transdermal delivery of the topical anaesthetic. We hypothesise that our novel microneedle design can thus directly deliver lignocaine to the skin nociceptor area, bypassing the thick stratum corneum layer, resulting in a faster onset of action and more effective dermal analgesia than the 5% EMLA dermal sham patch.

Hence, this trial aims: i) to assess the systemic absorption and the pharmacokinetic (PK) parameters of lignocaine in the blood circulation with this method of transdermal lignocaine delivery (Phase I); ii) To compare the efficacy of LEMAP with 5% EMLA sham dermal patch in pain relief during venepuncture based on VAS and skin algesimeter index assessments (Phase II) and iii) to evaluate the safety profile of LEMAP for topical analgesia in adult patients subjected to the routine venepuncture procedure (Phase I and Phase II). We hypothesise that, at one minute after venepuncture, participants receiving LEMAP will report lower pain than those receiving the EMLA sham patch on both co-primary outcomes. Specifically, the mean VAS will be lower with LEMAP than with control, and the mean SCAI will be lower with LEMAP than with control. We prespecify a patient-centred minimal clinically important difference (MCID) of 10 mm on the 0–100 mm VAS scale to guide interpretation and to underpin the Phase II sample size calculation.

## Methods

### Trial design and settings

This study is divided into two distinct phases:

a) Phase 1: Non-randomised single-centre open-label single group clinical trial to primarily assess the safety and tolerability of LEMAP in adult patients undergoing routine venepuncture procedure [Pharmacokinetic (PK) study].

b) Phase 2: A randomised, superiority, single-centre, double-blind, two-parallel-group, active-controlled clinical trial to assess the efficacy of LEMAP compared to 5% EMLA dermal sham patch [Pharmacodynamic (PD) study].

The SPRIT schedule of enrolment, intervention and assessment is represented in Fig 1.

The full trial schematics are summarised in Figs 2 and 3. Participant recruitment will commence in June 2024 and end in March 2026. The study is expected to be fully completed in August 2026.

### Recruitment

**Eligibility criteria and concomitant care.** Participants aged 18 years and above, requiring venepuncture for blood investigations at the Hospital Canselor Tuanku Muhriz (HCTM) ophthalmology outpatient clinic, Universiti Kebangsaan

| | TRIAL PERIOD | | | | | | |
|---|---|---|---|---|---|---|---|
| | **Phase I Trial** | | | | | | |
| | **Enrollment** | | **Post-intervention administration** | | | | **Close-out** |
| **TIMEPOINT**[b] | $-t_i$ to 0 | 0 | 30 mins | 60 mins | 90 mins | 120 mins | $t_5$ |
| **ENROLLMENT:** | | | | | | | |
| **Eligibility screen** | X | | | | | | |
| **Informed consent** | X | | | | | | |
| **INTERVENTION** | | | | | | | |
| **LEMAP administration** | | X | | | | | |
| **ASSESSMENTS:** | | | | | | | |
| **Venepuncture and blood sampling** | X | X | X | X | X | X | X |
| **Safety Assessment** | | | X | X | X | X | X |
| | **Phase II Trial** | | | | | | |
| | **Enrollment** | | **Post-randomization** | | | | **Close-out** |
| **TIMEPOINT**[b] | $-t_i$ to 0 | 0 | **30 minutes** | | | | $t_x$ |
| **ENROLLMENT:** | | | | | | | |
| **Eligibility screen** | X | | | | | | |
| **Informed consent** | X | | | | | | |
| **Documenting Baseline characteristics of study participants (e.g. baseline VAS and SCAI scores)** | X | | | | | | |
| **Randomization** | | X | | | | | |
| **INTERVENTION/ COMPARATOR:** | | | | | | | |
| **LEMAP administration** | | X | | | | | |
| **EMLA administration** | | X | | | | | |
| **ASSESSMENTS:** | | | | | | | |
| **Post-intervention VAS Score** | | | X | | | | |
| **Post-intervention SCAI assessment** | | | X | | | | |
| **Safety of intervention assessments** | | | X | | | | X |

**Fig 1. The SPIRIT schedule of enrolment, interventions and assessments for both Phase I and II trial stages (adapted from SPIRIT 2025 guideline).**

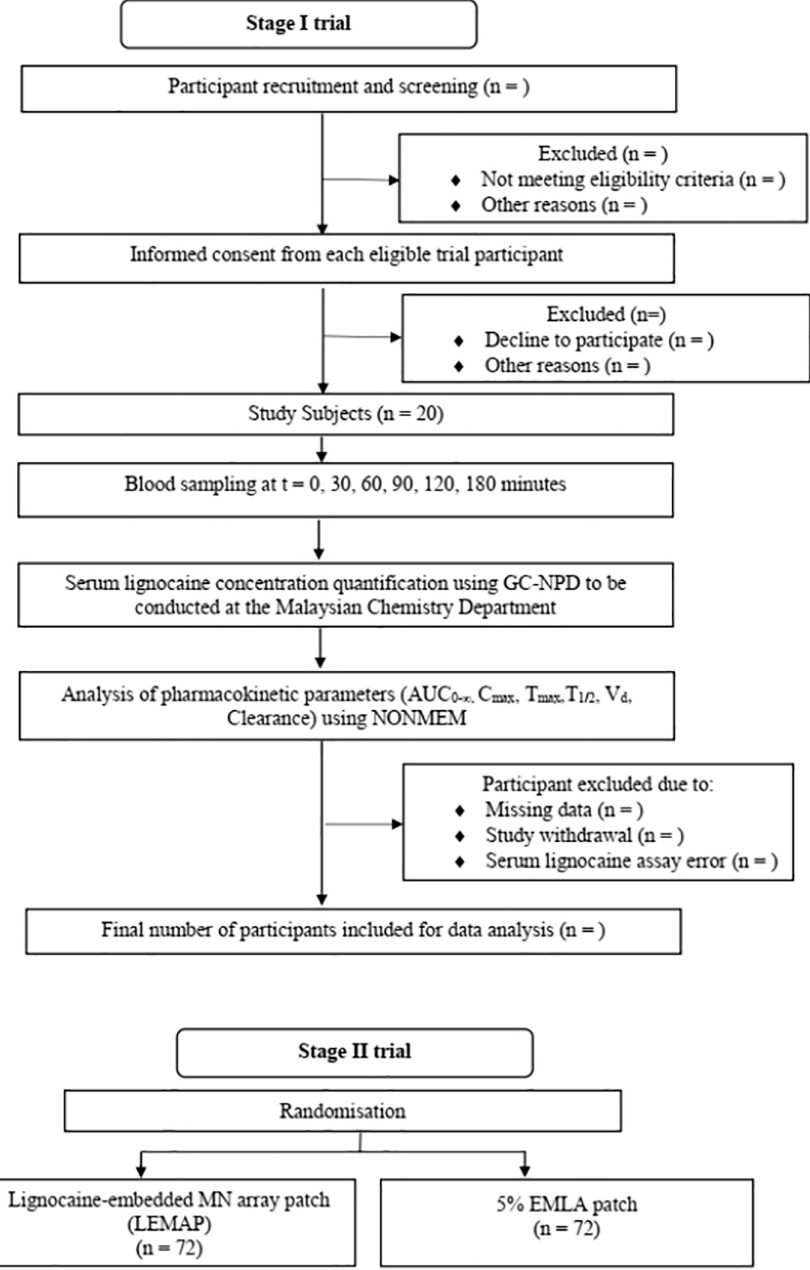

**Fig 2. The overall trial flowchart summarising the features of phase I and II trials.**

Malaysia (UKM), Kuala Lumpur, will be recruited. Those patients with the following will be excluded: i) prior history of sensitisation or allergy to lignocaine and other study materials (maltose, polyvinyl alcohol (PVA) and Polyethylene Terephthalate (PET)), ii) uncommunicative/deaf/mute patients, iii) exposure to analgesic usage within 24 hours before the venepuncture, iv) generalised skin disorder or rash, v) agitated and uncooperative patients, vi) current users of hypnotics and chronic pain relief medications, vii) those with psychiatric conditions or cognitive impairment viii) hepatic impairment, ix) users of CYP450 3A4, 3A5 or 1A2 inducers or inhibitors that may affect hepatic metabolism of the drug, x) failed first/ single attempt at venepuncture after the application of LEMAP or PET sham patch for the control arm.

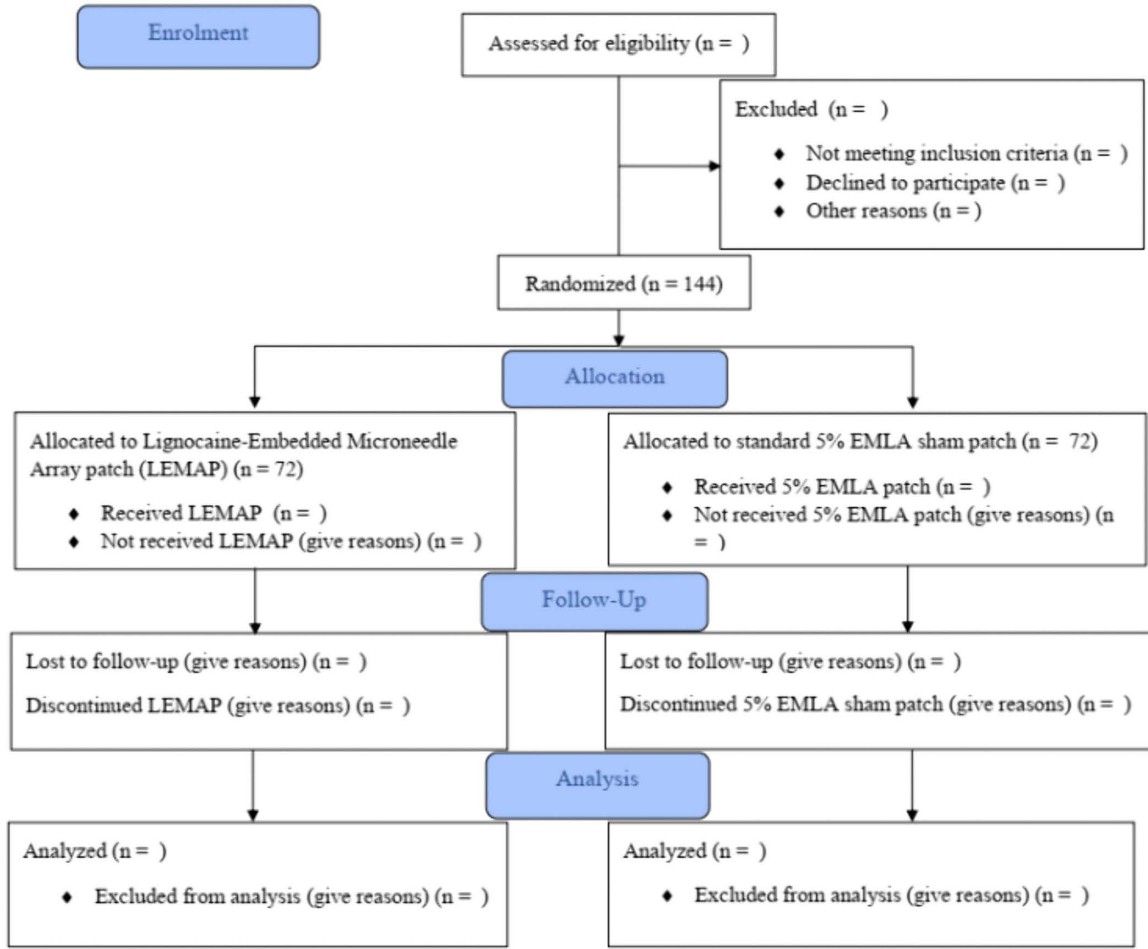

**Fig 3. The CONSORT diagram for the Stage II phase of the trial.**

As this trial involves a single application of the LEMAP or 5% EMLA™ dermal sham patch with immediate outcome assessment, no restrictions on concomitant care are necessary beyond the exclusion criteria already applied at screening. Participants are not expected to receive any concurrent therapies that would influence trial outcomes during the brief study period.

## Sample size calculation

**Phase I: Pharmacokinetic (PK) Study.** Due to the paucity of prior information, formal sample size calculation for stage I study (PK) based on power analysis cannot be carried out. Based on recommendations by Ogungbenro and Aarons [10] and Julious [11], the sample size is set at 20 subjects each for the PK study. Even though, it is recommended that the sample size can be minimally set at 12 subjects for a single-group pilot pharmacodynamic trial [11], the sample size is increased to 20 subjects since based on Julious's results (Fig 3 of the reference), the statistical asymptote is reached when the sample size is at least 20 subjects [11]. Hence, the addition of another subject will result in only a non-substantial gain in the precision of the parameter estimates when the sample size of 20 participants is reached.

**Phase II: Efficacy study.** For stage II of the trial (PD study), the sample size was calculated using Power and Sample Size (PS) Program version 3.1.6 (Vanderbilt University, Nashville, Tennessee, USA; 2018). We consider a 10-mm t VAS

difference on the 0–100 mm VAS scale between the intervention group as the MCID for acute procedural pain, based on the threshold approach reported in Olsen and colleagues in their systematic review [12] and to reduce the probability of type II error if the true treatment effect is actually closer to this conservative choice of MCID. Type I error and study power $(1 - \beta)$ were fixed at 0.05 and 0.80, respectively. We also assumed a moderate effect size (Cohen's $d = 0.5$) [13] and therefore the pooled standard deviation is assumed to be 20. The ratio of controls to cases is fixed at a 1:1 ratio. Based on these parameter values, the calculated sample size is 64 participants per group.

After accounting for a 10% drop-out rate, the final sample size is 72 participants per group ($n_{total} = 144$ participants) for phase II.

### Randomisation and blinding

**Randomisation to interventions and control arms.** Participants' eligibility will be verified by the primary investigators, medical officers and medically-qualified research assistants at the research site prior to participant recruitment. Each study personnel will receive the necessary training before the first patient is recruited. Block randomisation procedure with varying block size (permuted block) will be utilized to guarantee that both intervention groups will have an equal number of trial participants. This will be carried out by the trial statistician using the R package, blockrand version 1.50, which will be implemented on the R platform [14]. Eligible participants will be randomised after informed consent and all baseline measurements have been successfully concluded.

The list of generated random numbers will be used to allocate the study participants to either the intervention or the control arm. The allocation sequence generated will be kept in a password-protected document that is only accessible to the statistician to maintain allocation concealment. To further ensure the adequacy of allocation concealment, the randomisation code will not be revealed until the potential trial participants have been definitively enrolled into the trial, which will be after all baseline measurements are made and all eligibility criteria are deemed fulfilled by the study recruiters. In addition, allocation concealment is further safeguarded by ensuring the identity of the allotted treatment is only revealed to the interventionist (i.e., the person who will be administering the intervention) via secured telephone calls (central randomisation). Consecutive recruitments will be made until the final intended sample size is achieved.

**Blinding and emergency unblinding procedure.** For this study, the outcome assessors and care providers will be masked to the identity of interventions (double blinding). Only the statistician and interventionist/procedurist will be unmasked to the study interventions. Furthermore, unique ID code to indicate each treatment sequence assignment will be generated and utilised to ensure that the unintentional/intentional unmasking of one trial participant will not compromise the integrity of blinding for the rest of study participants. The primary unblinded trial persons (subjects, the statistician and the procedurist/interventionist) are instructed not to divulge the identity of the allotted treatments to other blinded trial personnel. The success of blinding will be determined by asking the blinded trial participants to guess the identity of the interventions received, which will then be compared with what would be anticipated by chance. Blinding indices such as James' Blinding Index or Bang's Blinding Index could also be calculated to objectively assess whether blinding has been successfully achieved in this trial [15,16].

If safety issues require the unblinding of the study participants, the requests should be initiated by the requesting clinicians. The Trial Management Group (TMG), which comprises grant holders and principal investigators, will adjudicate each unblinding request on a case-by-case basis. An independent study-site affiliated pharmacist who is granted access to the randomisation sequences will subsequently be consulted to complete the emergency unblinding.

### Intervention characteristics

**Physical descriptions of the microneedle array patch.** In this part, we will focus on the fabrication and implementation of the biodegradable microneedle array patch. Sugar compounds such as sucrose, trehalose, and

maltose have been experimented with as biodegradable matrix materials for microneedles. In particular, maltose per se is a carbohydrate that is widely acknowledged as a generally safe excipient material for drug delivery. Microneedle array patches (MAP) fabricated from maltose generally demonstrate strong mechanical properties, and as such, can facilitate perforation of skin and formation of micro-channels for transdermal drug delivery. Besides, for enabling the function as dissolvable MAP, maltose can rapidly dissolve in the dermal regions within minutes at body temperature, and is thus able to deliver therapeutic compounds rapidly, safely, and in an environmentally-friendly manner. The MAP that we propose as a prototype consists of two basic but essential structures, i.e., the microneedles and the substrate (baseplate). Both the microneedles and the patch backing are made from dissolvable materials, so there's no need to remove anything afterward. As the microneedles painlessly enter the upper layers of the skin, they begin to dissolve, gradually releasing the drug they carry in a controlled and sustained way. At the same time, the backing substrate also dissolves naturally, simplifying the application process. Furthermore, as the microneedle is only under 300 um in length, maximum penetration would only be limited to the epidermis-dermis intersection region. Hence, there is no possibility of the microneedles reaching the blood vessels, due to the restricted microneedle's lengths. Each microneedle array patch fabricated contains an estimated 12.5 mg of lignocaine. The lignocaine distribution will follow a topical mode of distribution. No significant systemic absorption of the drug is expected. An example of microneedle image and dermal application of LEMAP is given in Fig 4 (patent application number: UKM.IKB.800-4/1/5849; date of application: 16/04/2024).

Each LEMAP contains 225 microneedles per cm². The tip radius is 2 μm and the base radius is 50 μm. Lignocaine load is assumed to be uniformly distributed within each microneedle. The total lignocaine volumes per microneedle before application and after application are approximated using the conical volume formulae, and with r as the base radius, h as the microneedle height before LEMAP application and after LEMAP application. The maltose matrix volume was considered as negligible for lignocaine dose calculation. Post-application, about 20 percent of the initial lignocaine typically remains undissolved within the microneedle, so approximately 80 percent is potentially delivered to the skin. This geometric approximation is used only to standardise reporting of dose delivery since pharmacokinetic sampling in our Phase I trial remains the determinant of systemic exposure.

**Lignocaine characterization and formulation.** Lignocaine (Product Code: HY-B0185; CAS No. 137-58-6) was purchased from MedChemExpress (MCE), USA, in crystalline powder form. According to the Certificate of Analysis (Batch No: 287934), the product has a certified purity of 99.85% as determined by high performance liquid chromatography (HPLC). The compound was verified to have a moisture content of 0.26% (Loss on Drying) and 0.90% residue on ignition. The assay was calculated at 98.70% using standard gravimetric correction. The product was stored at 4°C in sealed amber glass containers to maintain stability and prevent degradation induced by light or moisture. All manufacturing and quality control processes adhere to ISO 9001:2015 certification standards. Lignocaine was handled following its Material Safety Data Sheet (MSDS) classification, which designates lignocaine as harmful if swallowed and potentially irritating to skin, eyes, and respiratory tract (Globally Harmonized System (GHS) Categories: Acute Toxicity: 4; Cutaneous Irritation: 2; Eye Irritation: 2A; Specific Target Organ Toxicity Upon Single Exposure (STOT SE): Category 3).

**Preparation of microneedle matrix mix for the dissolving microneedle.**

I. To prepare the calcium ion cross-linked alginate/sugar (CaCl2/SodiumAlg-sugar) composites, sodium alginate powder will first be dissolved in deionised (DI) water and stirred in a water bath until a homogeneous solution is gained.

II. Then, the $CaCl_2$ solution will be slowly added with rapid mixing to cross-link the alginate

III. To enhance the mechanical properties of composite microneedles, maltose will be added simultaneously into the mix solution to form a precursor for the preparation of paste for the fabrication of drug-loaded microneedles (Fig 5).

**Fabrication of dissolving microneedle array patch.**

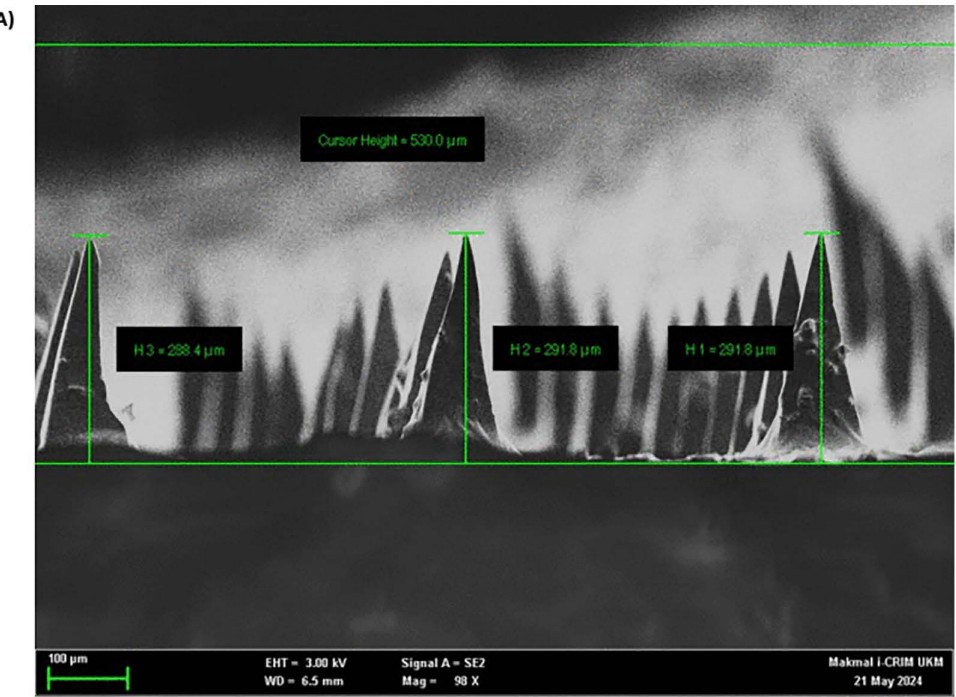

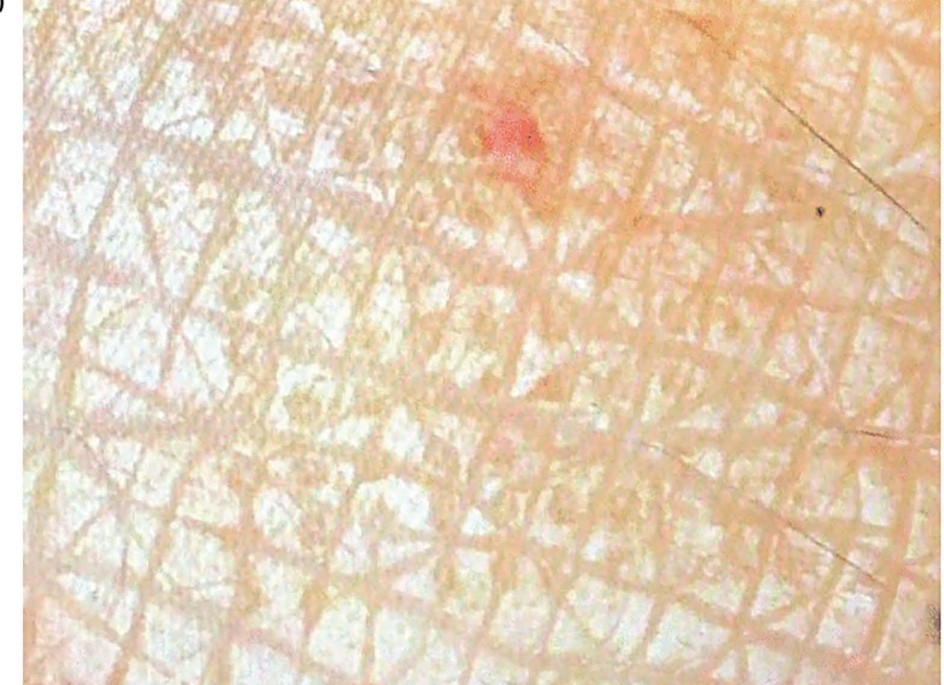

**Fig 4. Scanning electron microscopy (SEM) image of the microneedle patch and its dermal application effects.** (A) Scanning electron microscopy (SEM) image of microneedle patch, showing conical microstructures with uniform geometry (average height range: 288.4–291.8 µm; total array thickness: 530 µm). (B) Image of skin following LEMAP application, showing a single erythematous puncture mark at the application site, indicating skin penetration.

I. A casting process will be used to fabricate CaCl2/Alg-sugar composite microneedles. Firstly, the microneedle matrix loaded with the anaesthetic drug is poured onto the mould.

II. Secondly, the sample will be centrifuged to fill up the porous container of the microneedle mould to form a microneedle.

III. Thirdly, the preformed drug (lignocaine)-loaded microneedle will be heated in an oven forming hardening to become a final microneedle array patch as shown in a schematic diagram in Fig 5.

IV. The microneedles will be sterilized for sanitation by using gamma irradiation before packaging.

The microneedle patch will be made at the laboratory in the Institute of Microelectronics and Nanotechnology (IMEN), Universiti Kebangsaan Malaysia. Fig 5 summarise the microneedle fabrication steps. The schematics of LEMAP embedment process are given in Fig 6.

### Intervention

**Stage I trial.** For this stage I trial, to assess the safety and tolerability of 12.5 mg LEMAP, 20 healthy adult patients (10 males and 10 females) will be recruited. Pre-treatment fasting is not required for the participants. Each potential participant will be screened for study eligibility based on our pre-specified inclusion and exclusion criteria. An interim abridged medical history will be taken from each participant and their list of medications will be reviewed. Vital signs (systolic and diastolic blood pressures, oral temperature, pulse and respiratory rates) will be taken and targeted clinical examinations will be performed by the medical officers to assess the overall health of the participants.

First, an intravenous cannula will be placed at the dorsum of the hand, and the routine bloods taken for investigations, plus approximately 3.0 ml venous blood samples, will then be withdrawn at t = 0. Then the 12.5 mg lignocaine-impregnated microneedles will be applied to the antecubital fossa, after which blood will be drawn from the earlier inserted intravenous

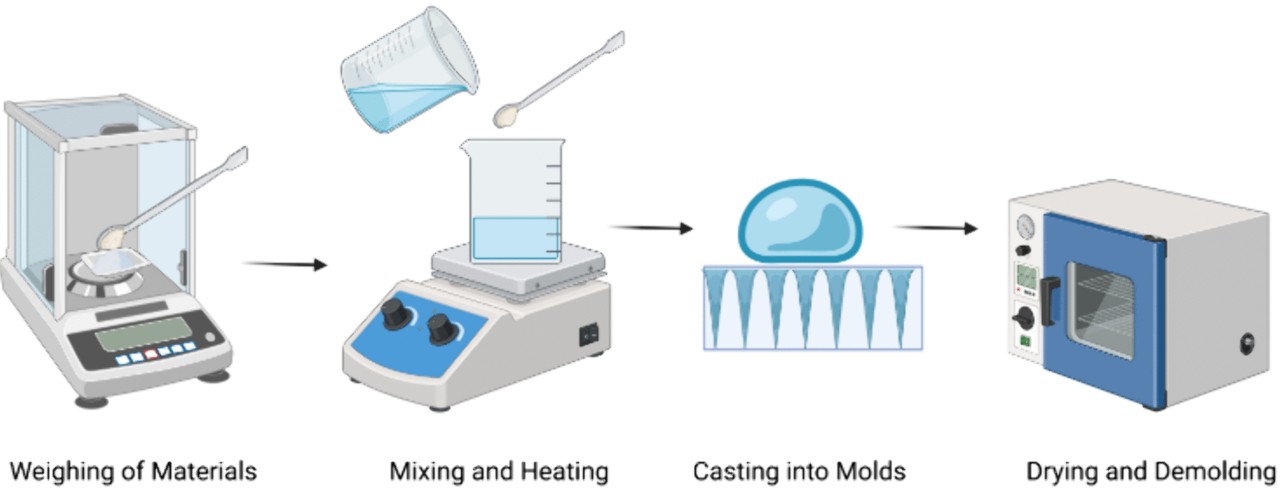

**Fig 5. Fabrication steps in making dissolvable microneedles: weighing materials, mixing and heating, pouring into molds, then drying and removing the microneedles.**

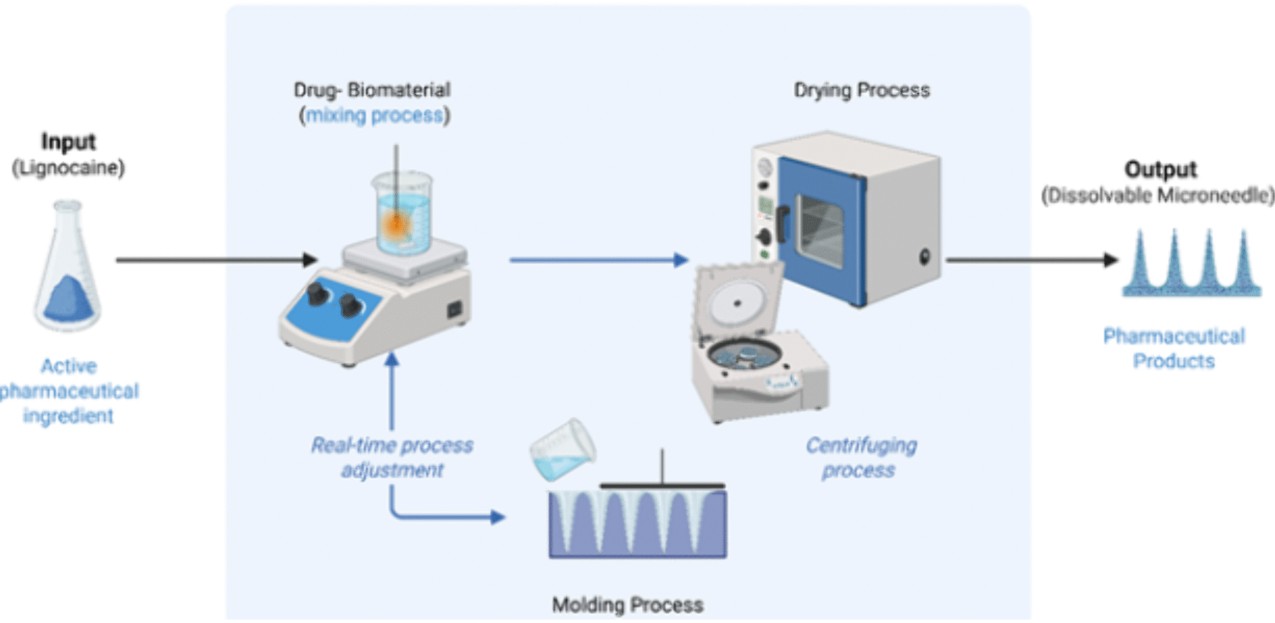

**Fig 6. The process of lignocaine embedment within the LEMAP structural design.**

cannula at t = 30, 60, 90, 120, 180 minutes and collected into separate 3.5-ml of plastic blood collection tube with accelerator & separator gel (BD™, New Jersey, USA). Heparinized saline will be periodically infused to ensure that the cannula lumen remains patent throughout the sampling periods. The blood samples will then be sent to the Malaysian Chemistry Department for lignocaine concentration measurements using the validated Gas Chromatography Nitrogen Phosphorus Detector (GC-NPD) technique.

**Determination of serum lignocaine concentration using Gas Chromatography Nitrogen Phosphorus Detector (GC-NPD).** One (1) mL of the blood will be taken from the collection tube and alkalinized with NaOH solution of pH 12. The internal standard, Methaqualone and the organic solvent, Chlorobutane will then be added to the mixture. The mixture will be subsequently mixed using a roller mixer and centrifuged to extract the organic layer, which will be concentrated from the partition. A clean-up solution, hexane-ethanol, is added to the sample mixture, vortexed and centrifuged. The bottom organic layer is transferred into another tube and evaporated to complete dryness under nitrogen gas flow at room temperature. The residue is then reconstituted using absolute ethanol prior to loading into the GC-NPD system.

Lignocaine in the blood matrix will be spiked based on the levels below or within the therapeutic range. The resolution and the peak of the lignocaine in the chromatograph will be recorded. In normal practice, lignocaine at the amount of 0.5 parts per million (ppm) will be spiked and a lower amount of lignocaine (0.3 ppm) will be used for quality control, which are based on the previous recommendations by Winek et al (Lignocaine: Therapeutic: 1.5-5.0 ppm; Toxic: 7–20 ppm; Lethal: > 25 ppm) [17]. For calibration, a 1-point calibration to estimate serum lignocaine concentration will be used. A series of 1-point calibrations will also be carried out whenever serum lignocaine concentration exceeds the therapeutic range.

**Post-intervention monitoring and pharmacokinetics data analysis.** Upon completing the collection of the last blood sample at t = 180 minutes, the participants will be further monitored for any adverse events (AEs) such as redness, pain, itchiness, blistering – (local reactions); and light-headedness, euphoria, tinnitus, diplopia – (systemic reactions) and any serious adverse events (SAEs) or suspected unexpected serious adverse reactions (SUSARs) for up to 48 hours via telephone calls.

The pharmacokinetic data will first be summarized in mean/standard deviation or median/interquartile range for continuous data and count and percentage for categorical data. The pharmacokinetic parameters ($AUC_{inf}$, $AUC_t$, $C_{max}$, $C_{min}$, $t_{max}$, $t_{1/2}$, volume of distribution ($V_d$), Clearance (Cl)) of lignocaine will be evaluated using blood samples obtained at times t = 0, 30, 60, 90, 120, and 180 minutes after the application of LEMAP. The intraindividual and interindividual variations of the pharmacokinetic parameters will be evaluated using the coefficient of variation (CV) and these will be classified as low (CV ≤ 10%), moderate (CV ≈ 25%) and high (CV > 40%) [18]. The pharmacokinetic data will be analysed using the non-linear mixed effect models based on the two-compartmental model, which will be implemented on NONMEM® version VI (Icon Development Solutions, Ellicott City, Maryland, USA). The influence of clinically relevant covariates such as participants' age, gender, BMI and others on pharmacokinetic parameters will be evaluated in a stepwise fashion. First-order conditional likelihood (FOCE INTER on NONMEM) will be used to fit the data, and model selection will be carried out using the likelihood ratio test, the estimates of pharmacokinetic parameters and their 95% confidence intervals, and goodness-of-fit measures.

**Stage II efficacy trial.** For the stage II trial, a different set of 142 participants to stage I participants will be recruited. Before the LEMAP application, relevant clinic-demographic profiles (age, gender, ethnicity, anthropometric measurements, presence of comorbidities) will be recorded and entered in the case report forms (CRFs) that are specifically designed for this study. The comparison of pharmacodynamic properties (i.e., efficacy) between 12.5 mg lignocaine delivered through LEMAP and 5% EMLA™ dermal sham patch containing one finger-tip unit (FTU = 0.5g) of 12.5 mg lignocaine and 12.5 mg prilocaine will be assessed via VAS score and skin algesimeter index for the pain induced by venepuncture.

The window period given to lignocaine for it to be effective will be based on the usual clinical practice observation, where it is usually applied for 30 minutes prior to venepuncture. The rationale behind it is due to logistical issues and for the daily clinic operational convenience. Nevertheless, in a busy clinic setting, the application time may sometimes be even shortened to 15 minutes for some anaesthetic effect. Thus, we postulate that, with the LEMAP, the time to onset of action for lignocaine could be greatly reduced, resulting in a much more effective pain reduction.

**Intervention group (LEMAP).** The total size of the patch is 1 cm (length) x 1 cm (width) x 730 µm (total patch thickness). Hence, the administrator of interventions (procedurist) will identify and draw a grid of 1 cm × 1 cm at the antecubital fossa, which will serve as an ideal site for venepuncture. Although LEMAP fully dissolves within approximately 2 minutes, the procedurist will keep the patch in place for 30 minutes to standardise application time, maintain blinding, and ensure adequate dermal uptake of lignocaine before cannulation. After 30 minutes, the medical personnel will perform the standard venepuncture procedure using a 21-gauge (G) hypodermic needle inserted into the vein beneath the patch.

**Control group (EMLA™ dermal sham patch).** For the participants allotted to the control group, one FTU of 5% EMLA™ cream will be applied and covered with a piece of PET patch to form a sham dermal patch. This will be applied and hold for 30 minutes similarly on the pre-marked area on the antecubital fossa. Again, standard venepuncture procedure will be performed after 30 minutes as described above.

### Trial outcomes

**Primary endpoint.** The Phase II trial has two co-primary endpoints. The first one is the Visual Analogue Scale (VAS) score, which will be measured on a continuous scale (range 0–100) using a Med-05–100 VAS Pain Scale ruler (Schlenker Enterprises Ltd, Lombard, USA) with a 0–100 mm slider. The measurement will be made one minute after venepuncture is performed, post-application, with either LEMAP or EMLA™ sham patch application by a trained medical officer with

at least 4 years of clinical experience following completion of the internship programme. A higher VAS score indicates greater intensity or degree of pain.

For VAS Pain Scale ruler calibration and standardisation, we will use a 0–100 mm slider VAS Pain Scale ruler of fixed length. Before first use and weekly thereafter, the research team will verify the physical scale length against a well-calibrated and standardized steel ruler and confirm the 0 mm and 100 mm markings on the VAS ruler align 0 mm and 100 mm markings on the reference ruler. VAS ruler that fails this check will be replaced. The participant will set the slider unaided, the trained outcome assessor (medical officer) will read the value from the reverse side.

The second co-primary endpoint is the skin conductance algesimeter index (SCAI). This will be measured in microSiemens per second (µS/s) and will be obtained using PainMonitor™ (Med-Storm Innovation AS, Oslo, Norway). The device consists of a sensor will be attached to the hypothenar eminence of the hand of the limb not subjected to venepuncture. The skin will be cleaned with alcohol swab and allowed to air dry before PainMonitor™'s sensor is attached. The single-use sensor will be applied with firm, even contact according to the manufacturer's instructions. The wrist and hand will be supported to minimise motion. The same limb and site will be used across participants whenever feasible.

Similar to VAS measurement, SCAI scores will be obtained at one minute after venepuncture post-LEMAP or EMLA™ dermal sham patch. Higher SCAI scores also indicate greater pain intensity. The SCAI measurement will be conducted by the same medical officer who has undergone training with the manufacturer for at least ten SCAI recording sessions.

For SCAI calibration, the operator will perform the PainMonitor™'s built-in quality check, confirm stable baseline conductance and acceptable signal quality and document ambient temperature and obvious sources of electrical or motion artefact. A baseline of at least 60 seconds will be recorded before venepuncture with the participant resting quietly. The post-venepuncture SCAI value obtained at one minute will be the primary time point, as prespecified above. If artefact is detected during this 60-second period, the operator will mark the trace and repeat a one-minute acquisition once the artefact resolves, without revealing treatment allocation. The same device model will be used throughout and operators will follow a standard operating procedure that includes periodic retraining and inter-operator checks.

**Secondary endpoints.** No secondary endpoint will be measured

**Safety assessment.** The following definitions, grading framework, reporting timelines, and oversight apply to both Phase I and Phase II. We define adverse events (AE) as "an abnormal sign, symptom, laboratory test, syndromic combination of such abnormalities, untoward or unplanned occurrence (e.g., accident), or any unexpected deterioration of concurrent illness" [19]. For serious AE (SAE), this is defined as "adverse events that result in the following outcomes: 1) death; 2) life-threatening AEs; 3) inpatient hospitalization or prolongation of existing hospitalization; 4) a persistence of significant incapacity or substantial disruption of the ability to conduct normal life functions, or a congenital anomaly or birth defect [20].

We classify the likelihood of AEs/ SAEs (unrelated, possible, probable, definite) based on Naranjo et al. classification [21]. All AE/SAE will be recorded and graded based on the Common Terminology Criteria for Adverse Events (CTCAE) Version 5 and US FDA's Toxicity Grading Scale Healthy Adults and Adolescents Volunteers Enrolled in Preventive Vaccine Clinical Trials [22,23]. All AEs or SAEs can be classified into local skin reaction (pain, erythema, ecchymosis, swelling, itchiness, tenderness) or systemic reaction (fever, irritability, tiredness, anorexia, vomiting, tachycardia, seizure, hypotension).

All AEs will be recorded on the CRFs. The detailed characteristics, the time and dates of onset and disappearance, and the severity of AEs will be included in the CRFs. The study investigators will assess each participant experiencing AEs and they will receive appropriate treatments accordingly. The relationships between AEs and LEMAP will be evaluated by the investigators and classified as either unrelated, possible, probable or definite based on Naranjo et al. classification. AEs are considered unexpected AEs when the AEs are not previously observed and not reported in the Investigator's Brochure or standard lignocaine package insert. Any incidence of AEs or SAEs classified as possibly, probably and definitely linked to LEMAP will be monitored until the AEs/SAEs resolution is complete or the Investigator deems that the AEs or SAEs have become stable or irrevocable.

 

All AEs of grade 3 and above will be reported to the UKM Research Ethics Committee (JEPUKM) within 5 business days. All SAEs (including Serious Unexpected Suspected Adverse Events (SUSARs)) will be reported within 24 hours of occurrence (expedited reporting) to the JEPUKM. If AEs/ SAEs occur or are still ongoing by the end of the study period, the study participants will still be continuously followed up until complete resolution of AEs/ SAEs which will take the following form: 1) additional participant visit to the trial centre/hospital; 2) telephone calls to the subjects; 3) additional reporting in the form of letters from the treating physicians.

Participant enrolment, intervention allocation and administration will be stopped if one of the following occurs (study halting criteria):

a) Death related to lignocaine-impregnated MN patch

b) Any participant experiences bronchospasm, laryngospasms or anaphylaxis within 24 hours post lignocaine-impregnated MN patch

c) Any SAE related to LEMAP

d) Any AE of grade 3 and above or any SAE that cannot obviously be attributed to other causes

e) Any study participant who develops abscess, ulceration or erosion at the site(s) of LEMAP

The study halting criteria above apply to both Phase I and Phase II. To ensure the independence of safety monitoring, all recorded safety data will be reviewed by JEPUKM which serves as the UKM Data Safety Monitoring Board (UKM-DSMB).

## Participant recruitment and retention strategy and recruitment monitoring

Eligible participants, specifically ophthalmology patients scheduled for eye procedures or follow-up visits at the ophthalmology outpatient clinic, will be identified through the weekly clinic appointment lists. A medically qualified research assistant will verify each patient's eligibility for the trial and will document this confirmation in the patient's medical records following successful consent. To support patient recruitment, medical officers at the HCTM ophthalmology outpatient clinics will be requested to notify the research team if potentially eligible participants are identified during regular outpatient consultations.

Recruitment progress will be tracked using a simple traffic light system:

• **Green**: Twelve or more participants are recruited and randomised each month, with an attrition rate below 10 percent.

• **Yellow**: Between nine and 11 participants are recruited and randomised each month, with an attrition rate below 10 percent. Strategies will be implemented to increase the recruitment rate to reach the green level.

• **Red**: Fewer than nine participants are recruited and randomised per month, with an attrition rate below 10 percent, or the three-month rolling average remains below nine participants despite mitigation. No effective strategies are available to improve recruitment to the green level sustainably.

If recruitment falls into the red category, the trial will be terminated. If recruitment remains in the yellow category for those two consecutive months, the trial will also be stopped early, following a two-month observation period.

As all trial activities, including eligibility assessment, informed consent, randomization, intervention, and outcome assessment, are conducted during a single clinic visit, the likelihood of participant drop-out or incomplete follow-up is minimal. In the event of protocol deviations or early discontinuation after randomization, primary outcome data, namely VAS and SCAI measurements and adverse event reporting, will still be collected where feasible, provided venepuncture has taken place.

The progress of participant recruitment will be overseen by the Trial Monitoring Group (TMG), which consists of the principal investigator and the trial statistician. The TMG will provide recommendations on the continuation of the trial, which will be communicated and discussed with the funder on a bimonthly basis.

## Ethical issues and dissemination of results

This study will be conducted in accordance with the principles of ethics in human research as stipulated by the Declaration of Helsinki (18th World Medical Association General Assembly, 1964), the Good Clinical Practice (GCP) guidelines, and ISO14155:2020 Clinical Investigation for Medical Devices for Human Subjects. Ethical clearance has been approved by the UKM Research Ethics Committee (JEPUKM) on 12th December 2022 (reference no: UKM/PPI/111/8/JEP-2022–736).

Voluntary written informed consent will be obtained from all study participants before study participation and randomization by trained medically-qualified trial research assistants (participant recruiters). The participants will be made aware that their participation is completely voluntary and they can withdraw from the study at any time point. The research participants will also be notified that their decision to withdraw from the study will not jeopardise their current or subsequent treatments and healthcare services. To ensure the confidentiality of patient information, each participant will be assigned an anonymous research ID code that will be used for data storage and analysis. The data will solely be made available to the research team members and access to the storage may only be granted by the principal investigators.

To enhance the transparency of reporting, the trial has been registered on the Clinical Trials Registry platform (https://clinicaltrials.gov/study/NCT05694858; Trial ID: NCT05694858, Date of registration: 19/01/2023, S2 Table). The full trial protocol has been made available in the same trial registries, prepared and revised according to the 2025 Standard Protocol Items: Recommendations for Interventional Trials (SPIRIT 2025) statement (S1 Table for SPIRIT 2025 item checklist) [24]. Any future protocol modifications will be submitted first to the JEPUKM for approval and the list of protocol changes will be made available to the public via both the Clinical Trials and NMRR registries.

The trial data will be shared on Harvard Dataverse (https://dataverse.harvard.edu) to ensure the transparent dissemination of study findings and adherence to recommendations made by leading medical journal editors for the future publication of this research. The deposit will include raw and processed VAS and SCAI outcome data, de-identified analysis datasets, PK concentration–time data and derived outputs, blank CRFs, screening and randomisation logs, protocol deviation log, and a complete variable dictionary and codebook. All shared files will be released under a Creative Commons Attribution 4.0 International licence.

## Statistical analysis plan

Data analysis will be performed using Statistical Package for Social Science (SPSS™) (IBM Corp. Released 2020. IBM Statistics for Windows, Version 27.0, Armonk, NY: IB Corp) and R version 4.5.1 (The Great Square Root. R Core Team, Vienna, Austria, Released on 13/06/2025). Our primary analysis will be based on the intention-to-treat (ITT) principle by which all trial participants will be analysed according to their original intended treatment assignment. For missing observations, we will use the multiple imputation method to fill in the missing data, assuming the missing at random (MAR) mechanism if the proportion of missing data per variable is more than 5% [25]. Specifically, we will employ multiple imputation by chained equations (MICE) with a predictive mean matching algorithm to impute missing outcome data [26]. We will generate 20 multiply-imputed datasets (increasing the number of imputations if needed for stability) and include baseline VAS and SCAI scores as auxiliary variables in the imputation model. This approach ensures that all randomized participants are analysed and that the imputation draws on relevant pre-pain measurements to make the MAR assumption as plausible as possible [27].

To assess departures from MAR (i.e., Missing Not At Random (MNAR)), we will perform pattern-mixture models with delta adjustment and a tipping-point analysis over clinically-plausible offset [27]. We will explore a range of $\delta$ values for the pain outcomes (e.g., up to ±1.0 VAS point and up to ± 10% in SCAI units) to represent plausible worse-outcome scenarios for those with missing data. The results will be re-evaluated under each $\delta$ and we will identify the tipping point at which the statistical significance of the between-group difference changes (from $p < 0.05$ to $p \geq 0.05$, or vice versa). This will illustrate how robust the primary outcome is to departures from the MAR assumption. No interim analyses will be conducted.

The differences in terms of VAS and SCAI (trial endpoints) between the groups will be statistically assessed using an independent t-test if the parametric assumption is met (i.e., the data are normally distributed) or the Mann-Whitney test if the data are non-normally distributed. Family-wise two-sided α of 0.05 will be controlled by requiring significance on both co-primary endpoints (VAS and SCAI). Each endpoint will be tested at two-sided α of 0.05. This conjunctive rule provides strong control of the Type I error in line with the US Food and Drug Administration's (US-FDA) guidance on multiple endpoints for confirmatory trials [28].

To control and adjust the effects of clinically-justified pre-specified covariates (confounders) such as baseline pain scores (baseline VAS and baseline SCAI), body mass index (BMI), gender, multiple linear regression (MLR) analysis will be used. We will not use automated or data-driven variable selection (no stepwise/AIC/BIC or purposeful selection). In addition, if any other measured baseline covariate shows a standardised difference >10% between arms, it will be added under this pre-specified rule to mitigate chance imbalance. Baseline balance will be summarized using standardized differences, but significance testing of baseline characteristics will not guide model specification. Effect modification will be assessed by creating and assessing the significance of the statistical interaction terms. The goodness of fit of the model will be evaluated using the coefficient of multiple determination, .

Model assumptions (linearity, independence, normality, and homoscedasticity of residuals) will be assessed using studentised residual vs predicted values scatter plots and Durbin Watson statistics. Box-Cox transformation with a carefully selected exponent (lambda) will be employed in the presence of skewness in the outcome variables. The presence of influential observations will be suspected in the presence of large leverage (extreme value in the x space) and residual values (extreme value in the y space) for any outlying observations. This will be further confirmed using influential diagnostic measures such as Cook's distance [29], dfFITS and dfBeta [30]. If the MLR model diagnostics indicate important assumption violations that are unresolved by transformation, we will proceed with two pre-specified alternatives. First, we will fit quantile regression at the median using the same covariates and report an adjusted median difference with bootstrap confidence intervals (CIs). Second, we will also fit a generalised additive model (GAM) with penalised splines for continuous covariates while retaining a parametric treatment term. The significance threshold will be set at 0.05 and 95% CIs will be presented for each effect estimate.

To assess the robustness of our findings, several additional sensitivity analyses will be conducted. We will compare the results from the primary multiple imputation approach (assuming the missing data are missing at random, or MAR) with those from complete case analysis. Alternative imputation strategies will be explored by varying the number of imputations, modifying the imputation model (e.g., auxiliary variables, interaction terms), and examining their effects on estimates. A per-protocol analysis will complement the ITT analysis by including only participants who fully adhered to treatment. Besides, in a separate sensitivity analysis, we will also fit the primary model after excluding observations identified as influential by standard diagnostics (Cook's distance, DFFITS, and DFBETAS) and compare the estimates with those obtained using the full dataset. To assess sensitivity to model specification, we will compare the primary regression model with alternatives based on clinical judgement, stepwise selection (AIC/BIC), penalised regression methods, such as the least absolute shrinkage and selection operator (LASSO) method, and between GAM and quantile regression if the MLR assumptions are violated. For penalised regression methods (e.g., LASSO), they will be used for exploratory purposes only to check model stability and will not change the primary efficacy inference. The treatment indicator will be included in all penalised models and not penalised and the findings will be reported descriptively. Consistent results across these sensitivity analyses will support the robustness of our trial findings and conclusion.

## Data management

The participant data on the case report form (CRF) will be anonymised and entered into a password-secure electronic database that is only accessible to the trial investigators. The data will be securely stored for 10 years following the trial completion at a secure site, after which the data will be permanently deleted. Data sharing will not be granted until the publication of trial results.

To further ensure the data integrity, all data entry procedure will follow a standardised coding system and will be performed by trained trial research assistants. To promote data accuracy, 10% of the entries will undergo double data entry and cross-verification. Built-in validation rules, including range and consistency checks, will be implemented to identify out-of-range or illogical values in real time. Regular data audits and cleaning procedures will be conducted by the trial research assistants overseen by the trial statistician throughout the trial to ensure data integrity. All data will be backed up routinely and stored in compliance with the UKM and the Malaysian National Pharmaceutical Regulatory Agency (NPRA)'s data protection guidelines.

## Trial management and oversight

The trial will be conducted in accordance with the Good Clinical Principle (GCP) guideline and the Malaysian NPRA's clinical trial guideline. This study is sponsored by the Malaysian Ministry of Science, Technology and Innovation (MOSTI) under the TED2 grant scheme which will also be responsible for monitoring the trial's overall progress. The Trial Monitoring Group (TMG), which is overseen by the principal investigators and other trial members, will manage the overall conduct of the study.

## Data and safety monitoring board

The UKM Research Ethics Committee (JEPUKM), which constitutes anaesthesiologists, clinical triallists, statisticians and other clinicians, is responsible for the monitoring and evaluating the safety data generated in this trial. A 6-monthly safety and intervention efficacy report will be submitted to the JEPUKM, which will appraise the rates of AEs, SAEs, and SUSARs (including deaths) in trial participants. The DSMB will also review the data quality for each trial outcome and the overall trial progress. The findings from these six-monthly review processes will be used to determine the trial's continuation. Any major protocol amendments (e.g., changes in the trial eligibility criteria, addition or removal of new or old trial endpoints) made will be relayed to the JEPUKM within 30 days after approval by the TMG and before the new amended protocol is executed in the trial setting.

## Patient and public Involvement

Patients and the public do not explicitly participate in the development, design and writing-up of this trial protocol. However, relevant insights from independent non-trial patients with similar characteristics will be obtained to optimise the parameters of the fabricated LEMAP prior to conducting the first trial stage. All participants will be protected under private insurance coverage, which serves as a compensation mechanism in the event of any procedure-related adverse events, including death. No post-trial ancillary care is planned, as the intervention is limited to a one-time application for a single outpatient venepuncture for both phase I and II trials.

During the trial, participants will be asked to provide feedback on the trial conduct, monitoring, and documentation to the principal investigators via confidential means (paper-based or electronic survey forms, private phone calls, or emails). After the trial conclusion, the results and the identity of the intervention received will be shared and divulged to each trial participant.

## Trial status

The first trial participant for phase I was recruited on 17 February 2025 based on the trial protocol version 3.0. The previous trial protocol version was amended on 01 December 2023 to account for additional eligibility criteria. The second trial protocol was further amended on 14 December 2024 to include a more detailed explanation of the outcome measure assessment and the inclusion of eligibility criteria and investigators and an extension of the duration of ethical approval was granted on 30 December 2024 by the UKM Ethics Committee, which has also approved all amendments and revisions made to the trial protocol.

Phase I trial recruitment is expected to take eight months to complete and is thus anticipated to be finished on 16 October 2025. Phase II trial recruitment is scheduled to commence on 03 November 2025 and is expected to be completed by 11 December 2026, including data collection. After that, the trial database will be locked for data analyses, which will be until 31 January 2027. The manuscript containing the results from both trial phases is expected to be submitted for publication by 31/08/2027 at the latest, allowing for the dissemination of the trial findings. Besides, the results will also be deposited in the clinicaltrials.gov registry under the trial registration weblink (https://clinicaltrials.gov/study/NCT05694858, S2 Table). At the time of submission, no outcome data have been analysed and no unblinding has occurred. This article reports the pre-results trial protocol in accordance with SPIRIT 2025 guideline [24].

## Discussion

This trial will provide evidence on whether the transdermally-delivered lignocaine through LEMAP will result in more effective relief of pain induced by venepuncture among adults. For objective safety assessments, recent research indicates several methodological challenges. Pensado et al. demonstrated that the minimally invasive stratum corneum (SC) sampling was only adequately-powered to detect a 50% difference in the mean uptake and clearance parameters and pharmacodynamic response characteristics (measured using the area above the blanching effect curve (AAEC)) between two different doses of Betnovate® (0.1% w/w BMV) cream applied on the anterior aspects of the forearms of 12 individuals, whilst a larger sample size was required to detect a smaller effect size (20% difference) for the two applied doses [31]. In contrast, the skin blanching technique produced higher variability in the results and therefore could not discriminate much between the two different Betnovate® cream doses [31]. Other techniques that have been proposed for dermato-pharmacokinetic (DPK) assessments of topical drugs, such as the in vitro Franz Diffusion Cell, ex vivo tape stripping, in-vivo micro-dialysis and suction blister techniques and confocal laser microscopy, are also faced with serious methodological shortcomings (i.e., invasive procedure, absence of sensitive methods to analyze the analyte's concentration, technical variations in the tape removal) [32,33]. Consequently, the DPK properties of topically-delivered local anaesthetics could not be accurately evaluated, which is the principal reason why our research will focus more on the TDD system's safety profile evaluation through the quantification of the amount of lignocaine or 5% EMLA™ dermal patch entering the systemic circulation (classical bioavailability).

The following five main points can summarise the strengths and limitations of our trial design:

- This is the first clinical trial investigating the safety and efficacy of a novel analgesic drug-embedded dissolvable microneedle array patch that may enhance the transdermal delivery of lignocaine in adults subjected to routine venepuncture procedure in a typical hospital outpatient clinic setting.

- The phase II of the trial will be conducted as a randomised, double-blind and active-controlled trial which compares the efficacy of skin analgesia between the lignocaine-embedded microneedle design against the 5% EMLA dermal sham patch. The trial design will minimise the selection and confirmation bias and the results can be conveniently translated into changes in best practices in pain relief before venepuncture.

- A more objective pain assessment method based on the skin conductance algesimeter index (SCAI) will be used for primary trial endpoint evaluation. This may be more objective than the patient-specific subjectivity associated with Visual Analogue Scale (VAS) scoring and ranking.

- The main limitation of this trial is that it will be carried out in a single tertiary-care hospital, which may attenuate the generalisability of the results. A future multi-centre clinical trial is required to further verify the trial's results.

Nevertheless, we believe LEMAP may accelerate lignocaine's onset of action, and in the event of a successful trial, the current microneedle design may be further optimised and investigated in the paediatric population.

## Supporting information

**S1 Table.** **The filled-in SPIRIT 2025 Checklist.**
(DOCX)

**S2 Table.** **Components of the WHO Registration Data Set, extracted from the clinicaltrials.gov registry and adapted based on the 2025 SPIRIT guideline.**
(DOCX)

**S1 File.** **Full trial protocol (Transdermal Microneedle).**
(PDF)

## Acknowledgments

The authors would like to thank our trial collaborator, Dr Goh Chee Seong from Alnair Labs Corporation, Japan, for his kind advice and technical assistance in designing and fabricating the LEMAP. We also thank the contributions of our research assistants, Dr Ang Zheng Jiet, Dr Chieng Yew Wen, Dr Natalie Ling Sze Jane, Dr Aaron Phan Yong Hong, Dr Ching Xiu Wei and Dr Tan Ker Xin.

## Author contributions

**Conceptualization:** Muhammad Irfan Abdul Jalal, Fook-Choe Cheah.

**Formal analysis:** Muhammad Irfan Abdul Jalal.

**Funding acquisition:** Azlan Azrul Hamzah, Fook-Choe Cheah.

**Methodology:** Muhammad Irfan Abdul Jalal, Mun Yin Yen, Lam Chenshen, Mae-Lynn Catherine Bastion, Chua Xin Yun, Sharipah Salwa Abdul Razak, Sharmilah Kuppusami, Fook-Choe Cheah.

**Project administration:** Muhammad Irfan Abdul Jalal, Fook-Choe Cheah.

**Resources:** Arifah Syahirah Abdul Rahman, Mohd Eusoff Azizol Nashriby, Nurul Ashikin A Rahim, Poh Choon Ooi, Muhamad Ramdzan Buyong, Mohd Ambri Mohamed, Chang Fu Dee, Azlan Azrul Hamzah.

**Writing – original draft:** Muhammad Irfan Abdul Jalal, Mun Yin Yen, Lam Chenshen, Mae-Lynn Catherine Bastion, Chua Xin Yun, Sharipah Salwa Abdul Razak, Sharmilah Kuppusami, Arifah Syahirah Abdul Rahman, Mohd Eusoff Azizol Nashriby, Nurul Ashikin A Rahim, Poh Choon Ooi, Muhamad Ramdzan Buyong, Mohd Ambri Mohamed, Chang Fu Dee, Azlan Azrul Hamzah, Fook-Choe Cheah.

**Writing – review & editing:** Muhammad Irfan Abdul Jalal, Mun Yin Yen, Lam Chenshen, Mae-Lynn Catherine Bastion, Chua Xin Yun, Sharipah Salwa Abdul Razak, Sharmilah Kuppusami, Arifah Syahirah Abdul Rahman, Mohd Eusoff Azizol Nashriby, Nurul Ashikin A Rahim, Poh Choon Ooi, Muhamad Ramdzan Buyong, Mohd Ambri Mohamed, Chang Fu Dee, Azlan Azrul Hamzah, Fook-Choe Cheah.

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
