## [Decision Letter · Decision Letter 0]

8 Sep 2025

Dear Dr. Abdul Jalal,

Thank you for submitting your manuscript to PLOS ONE. After careful consideration, we feel that it has merit but does not fully meet PLOS ONE’s publication criteria as it currently stands. Therefore, we invite you to submit a revised version of the manuscript that addresses the points raised during the review process.

We look forward to receiving your revised manuscript.

Kind regards,

Jeffrey Pradeep Raj, MBBS.,MD.,PG Dip Fly Med, PG Dip Biostats, MBA

Academic Editor

PLOS ONE

Journal Requirements:

2. We note that the original protocol that you have uploaded as a Supporting Information file contains an institutional logo. As this logo is likely copyrighted, we ask that you please remove it from this file and upload an updated version upon resubmission.

3. Thank you for stating the following financial disclosure: [This study is supported by the Technology Development Fund 2 (TED2) Grant (Grant number: MOSTI.D (S) 600-4/19/15) funded by the Ministry of Science, Technology and Innovation (MOSTI), Malaysia and UKM Faculty of Medicine Grant (Grant number: FF-2021-401).]. 

6. Please include captions for your Supporting Information files at the end of your manuscript, and update any in-text citations to match accordingly. Please see our Supporting Information guidelines for more information: http://journals.plos.org/plosone/s/supporting-information .

Reviewers' comments:

Reviewer's Responses to Questions

**Comments to the Author**

1. Does the manuscript provide a valid rationale for the proposed study, with clearly identified and justified research questions?

Reviewer #1: Yes

Reviewer #2: Yes

2. Is the protocol technically sound and planned in a manner that will lead to a meaningful outcome and allow testing the stated hypotheses?

Reviewer #1: Partly

Reviewer #2: Yes

3. Is the methodology feasible and described in sufficient detail to allow the work to be replicable?

Reviewer #1: No

Reviewer #2: Yes

4. Have the authors described where all data underlying the findings will be made available when the study is complete?

Reviewer #1: Yes

Reviewer #2: Yes

5. Is the manuscript presented in an intelligible fashion and written in standard English?

Reviewer #1: Yes

Reviewer #2: Yes

You may also provide optional suggestions and comments to authors that they might find helpful in planning their study.

Reviewer #1: Thank you for the opportunity to review your study protocol. The topic is clinically important, and the two-stage design has clear potential to advance practice. Below is a point-by-point set of comments for your consideration.

1. Protocol prerequisites – The manuscript states that Phase I recruitment began on 17 February 2025 and is expected to finish by 16 October 2025, with Phase II due to start in November 2025 . Because participants are already being enrolled, please confirm that no outcome data have been analysed and explain how the manuscript continues to qualify as a pre-results “Study Protocol”.

2. Rationale & research questions – The introduction clearly identifies the gap (few adult data on microneedle anaesthesia for venepuncture) and proposes LEMAP as a solution. However, explicit directional hypotheses and a patient-centred minimal clinically important difference (MCID) are not stated. Please add these to sharpen the objectives.

3. Technical soundness of the design

3.1 Two-stage approach (Phase I PK safety; Phase II efficacy) is appropriate .

3.2 Sample size – The Phase II calculation assumes a “1-point VAS difference” with SD = 2 and Cohen’s d = 0.5 . Given that VAS is defined as 0–100 mm (not 0–10 points) elsewhere in the manuscript , the target effect appears to be ~10 mm. This is at the lower boundary of published MCIDs (≈9–13 mm) for acute pain; please justify or consider re-powering for ≥13–15 mm.

4. Blinding – LEMAP dissolves after 2 min while the EMLA sham stays occluded for 30 min, which may unmask allocation to clinicians. Describe tactile/visual masking steps or harmonise application times.

5. Methodological detail for replication – Device fabrication steps are outlined , but key specifications (needle density, tip radius, residual lignocaine after dissolution) are missing. Likewise, SCAI sensor placement/calibration and VAS ruler calibration should be elaborated to enable replication.

6. Statistical analysis plan – The SAP is detailed but internally inconsistent:

a. Primary comparisons use t/Mann-Whitney tests, then linear regression with covariate selection by multiple strategies. Mixing stepwise (AIC/BIC), purposeful selection and clinical judgement is overly flexible and can inflate false-positive rates. Specify a single, preferably pre-defined strategy—e.g., include only clinically justified pre-specified covariates plus any baseline imbalance >10 %.

b. Penalised methods (LASSO) are proposed as a sensitivity analysis; clarify whether shrinkage estimates will be purely exploratory or could alter primary inference.

c. Two co-primary endpoints (VAS, SCAI) require an α-adjustment, not mentioned.

d. Missing-data strategy. This is a single-visit study in which VAS and SCAI are recorded immediately after venepuncture, so missing outcome data should be rare. Unless you expect > 5 % missingness, multiple imputation may add unnecessary complexity. If you keep it, please pre-specify: (i) the algorithm (e.g., MICE with predictive mean matching), (ii) the number of imputations (≥ 20), (iii) the auxiliary variables to be included, and (iv) the δ values or shift parameters for any pattern-mixture sensitivity analyses.

e. Excluding influential observations conflicts with ITT; instead present them as sensitivity analyses.

f. The Abstract/Methods (lines 40-44) mention 142 participants = 71/arm, whereas the power section (lines 170-178) recalculates 72 per arm; 144 total. Kindly harmonise these figures throughout the manuscript.

7. Recruitment traffic-light plan is clear; nonetheless, the red-light threshold (< 9 participants / month) may prematurely terminate the trial if clinic flow fluctuates, consider adding a buffer period.

8. Stopping criteria are mentioned for Phase I PK, but Phase II lacks explicit adverse-event grading and reporting timelines. A brief statement (e.g., “All AEs graded per CTCAE v5.0 and reported within 24 h to the DSMB”) would close that gap.

9. Data-availability detail – The Harvard Dataverse statement is welcome, but it should pre-specify

a) the file types to be shared (e.g., raw and processed VAS/SCAI data, PK curves, blank CRFs),

b) a variable dictionary/codebook, and

c) the intended licence (e.g., CC-BY 4.0).

10. Competing-interest declaration – The methods section cites a patent application for the microneedle patch (UKM.IKB.800-4/1/5849, filed 16 Apr 2024), yet the “Competing interests” field states “None declared.” Please clarify whether any authors are inventors, hold a licence agreement, or have a financial stake, and update the declaration accordingly.

To round off an otherwise promising protocol, it would be helpful to clarify three items in prose. First, the stated “1-point” effect corresponds to roughly 10 mm on a 0–100 mm VAS; a short, patient-focused explanation of why this magnitude represents meaningful pain relief would solidify both the sample-size justification and the clinical relevance of the study.

Second, because the active patch dissolves in about two minutes whereas the control remains occluded for thirty, additional detail on masking procedures—such as matching occlusion times or using opaque dressings—would allow readers to judge the blinding integrity.

Lastly, full reproducibility would be achieved by specifying the final microneedle dimensions and sensor/ruler calibrations, and by tightening the statistical plan to one predefined covariate strategy and fixed imputation settings, with any case exclusions reserved for sensitivity analyses.

Reviewer #2: Dear Editor,

I write to submit my review on the protocol titled “The efficacy and safety of lignocaine-embedded dissolvable microneedle versus EMLA or topical analgesia in adults undergoing venepuncture: A Single-Centre, Parallel-Group, Double-Blind Randomised Clinical Trial Protocol in a Tertiary Care Setting”

Using single-centre, active-controlled, double-blind, randomised superiority trial, the protocol aims to investigate the safety and efficacy of a novel lignocaine-embedded transdermal microneedle array patch (LEMAP) in facilitating transcutaneous lignocaine delivery to reduce procedural-related pain in adults undergoing venepuncture in a tertiary-care outpatient clinic setting.

Here are my comments

Overall impression: The pharmacokinetic and statistical methodologies for Phase I and Phase II are on point and very detailed. It looks great with clearly defined endpoint measures, including highlighting the measurement scale and appropriate pharmacokinetic and statistical methods to address Phase I and Phase II objectives, respectively.

The sample size for the Phase I: Pharmacokinetic (PK) Study was properly justified, even though no formal power analysis for the PK study was done due to a lack of data or information on key sample size parameters, as articulated by the authors. Relevant references from previous studies were provided to justify the choice of sample size.

The power analysis for the phase II study was properly justified. However, authors must explain or provide justification or references regarding why they considered a 1-point VAS difference between the intervention group as the minimum detectable difference (MTD), as sample size is largely influenced in trial studies by (MDD). Note is MDD, not MTD.

For missing observations, the authors indicated that they will use the multiple imputation method to fill in the missing data, assuming the missing at random (MAR) mechanism. Since data are yet to be collected, it will be more prudent to include in the proposal other missingness mechanisms and how they will be addressed if they occur, as it is not guaranteed that the missingness mechanism will follow MAR. Why not MNAR, MCAR etc

Assuming MLR is not appropriate, include a non-parametric regression modelling alternative to MLR in the proposal, similar to what you did for the t-test by proposing the Mann-Whitney test

Line 521, kindly delete R2, is R-squared not R2. Kindly use proper equation editor to write it out well

**Do you want your identity to be public for this peer review?** For information about this choice, including consent withdrawal, please see our Privacy Policy

Reviewer #1: No

Reviewer #2: No

---

## [Author Response · Author response to Decision Letter 1]

13 Sep 2025

Appendix A

A) Reviewer 1: Point by Point Responses

1) Comment 1: Protocol prerequisites – The manuscript states that Phase I recruitment began on 17 February 2025 and is expected to finish by 16 October 2025, with Phase II due to start in November 2025. Because participants are already being enrolled, please confirm that no outcome data have been analysed and explain how the manuscript continues to qualify as a pre-result “Study Protocol”.

i) Author’s Response

We sincerely thank the reviewer for raising this point. We confirm that no outcome data have been analysed. No interim, exploratory, or final analyses of primary or secondary endpoints have been conducted. No unblinding has occurred. Monitoring has been limited to routine trial conduct checks such as screening logs, accrual counts, and safety oversight, which do not involve outcome analysis.

The submission qualifies as a pre-results Study Protocol because journals that publish protocols define “pre-results” by the absence of analysed outcomes and by the fact that recruitment or data collection is not yet complete. PLOS ONE’s reviewer guidance states that Study Protocols should not have generated results and recruitment or data collection should not be complete, while allowing feasibility or pilot information only when necessary for proof of principle [1]. Trials specifies that protocols are generally considered for proposed or ongoing studies before recruitment completes and may be considered up to before last patient last visit provided the manuscript clearly states the status at submission and explains any delay [2]. BMJ Open advises that studies that have not completed recruitment or baseline data collection should be submitted as study protocols [3]. These policies align with SPIRIT 2025, which governs the content and transparency of trial protocols and does not require that recruitment be entirely prospective for the protocol article itself, provided that all prespecified plans, governance, and reporting elements are fully described [4]. Together, these policies mean that a protocol can be submitted and published while recruitment is underway, so long as no outcome data have been analysed and the manuscript transparently reports trial status. [1-4]

For completeness, we also confirm prospective registration of our trial protocol and adherence to accepted standards. The trial was prospectively registered in a ICMJE-recognised public registry before the first participant was enrolled, in line with ICMJE policy and WHO standards on prospective registration. The registry identifier and registration date are reported in the manuscript. [5, 6]

To address your request for clarity within the manuscript, we will add the following statement in the “Trial status” section and in the cover letter.

ii) Manuscript changes (In Methodology, Trial Status Subsection)

“At the time of submission, no outcome data have been analysed and no unblinding has occurred. This article reports the pre-results trial protocol in accordance with SPIRIT 2025 guideline [28].”

iii) References Used in Responses

1. PLOS ONE. Reviewer guidelines. 2025.

2. Trials. Study protocols policy update. 30 January 2020. Also see Trials “Study protocol” submission guidelines, current version.

3. BMJ Open. Authors page. Guidance on protocol submissions for studies that have not completed recruitment or baseline data collection.

4. Hróbjartsson A, Boutron I, Hopewell S, Moher D, Schulz KF, Collins GS, et al. SPIRIT 2025 explanation and elaboration. BMJ. 2025;389:e081660. doi:10.1136/bmj-2024-081660.

5. International Committee of Medical Journal Editors. Clinical trial registration policy and FAQs.

6. World Health Organization. International standards for clinical trial registries. 2018.

2) Reviewer’s Comment 2: Rationale & research questions – The introduction clearly identifies the gap (few adult data on microneedle anaesthesia for venepuncture) and proposes LEMAP as a solution. However, explicit directional hypotheses and a patient-centred minimal clinically important difference (MCID) are not stated. Please add these to sharpen the objectives.

i) Author’s Response

Thank you for the suggestion. We have added explicit directional hypotheses and a patient-centred MCID to sharpen the objectives. The hypotheses state that LEMAP will reduce pain more than the EMLA sham patch on both co-primary outcomes (VAS and SCAI). We also prespecify a 10 mm MCID on the 0–100 mm VAS, which aligns with the value used for Phase II sample size and will guide interpretation. These additions appear at the end of the introduction section.

ii) Manuscript changes (In Introduction)

“We hypothesise that, at one minute after venepuncture, participants receiving LEMAP will report lower pain than those receiving the EMLA sham patch on both co-primary outcomes. Specifically, the mean VAS will be lower with LEMAP than with control, and the mean SCAI will be lower with LEMAP than with control. We prespecify a patient-centred minimal clinically important difference (MCID) of 10 mm on the 0-100 mm VAS scale to guide interpretation and to underpin the Phase II sample size calculation.” (Page 7, Lines 114-120)

3) Reviewer’s Comment 3 (specifically point 3.2): Sample size – The Phase II calculation assumes a “1-point VAS difference” with SD = 2 and Cohen’s d = 0.5. Given that VAS is defined as 0–100 mm (not 0–10 points) elsewhere in the manuscript, the target effect appears to be ~10 mm. This is at the lower boundary of published MCIDs (≈9–13 mm) for acute pain; please justify or consider re-powering for ≥13–15 mm.

i) Author’s Response

We thank the reviewer for this helpful comment. We agree that all pain outcomes should be expressed on the 0-100 mm VAS scale. We have corrected the units in the Phase II sample size paragraph and now state a 10 mm target difference with SD 20 mm, which corresponds to the same standardized effect size (Cohen’s d = 0.5) used previously. A 10 mm difference is consistent with the lower end of published minimal clinically important differences (MCIDs) for acute pain and has been repeatedly supported in emergency and perioperative studies [1-5]. Todd et al. reported about 13 mm as the smallest important change adults can perceive on a 100 mm VAS [1], with similar findings from Gallagher et al [2]. Kelly reported an MCID of 9 mm VAS in an emergency setting [3], and a postoperative study by Myles et al. triangulated an MCID of approximately 10 mm [4]. A systematic review found threshold-based MCIDs with a median of 10 mm VAS score across acute pain studies, based on the threshold method for absolute VAS change [5].

On this basis, 10 mm is a reasonable and conservative value to power a confirmatory trial. Re-powering to ≥13-15 mm would assume a larger effect than is necessary and could reduce the required sample size while increasing the risk of type II error if the true effect is closer to 10 mm. We therefore retain 10 mm (SD 20 mm) and have clarified the units in the manuscript with citations to the MCID literature.

ii) Manuscript changes (In Methodology, Sample Size Calculation Subsection, Under Phase II: Efficacy Study Sub-subsection)

“We consider a 10-mm point VAS difference on the 0-100 mm VAS scale between the intervention group as the MCID for acute procedural pain, based on the threshold approach reported in Olsen and colleagues in their systematic review [12] and to reduce the probability of type II error if the true treatment effect is actually closer to this conservative choice of MCID.” (Page 13, Lines 173-176)

iii) References Used in Responses

1) Todd KH, Funk KG, Funk JP, Bonacci R. Clinical significance of reported changes in pain severity. Ann Emerg Med. 1996;27(4):485-9.

2) Gallagher EJ, Liebman M, Bijur PE. Prospective validation of clinically important changes in pain severity measured on a visual analog scale. Ann Emerg Med. 2001;38(6):633-8.

3) Kelly AM. The minimum clinically significant difference in visual analogue scale pain score does not differ with severity of pain. Emerg Med J. 2001;18(3):205-7.

4) Myles PS, Myles DB, Galagher W, et al. Minimal clinically important difference for three quality of recovery scales after general anaesthesia. Br J Anaesth. 2017;118(3):424-9

5) Olsen MF, Bjerre E, Hansen MD, et al. Pain relief that matters to patients: Systematic review of MCID for acute pain measured on VAS/NRS. BMC Med. 2017;15:35. (reference 12 in the main manuscript text)

4) Reviewer’s Comment 4: Blinding – LEMAP dissolves after 2 min while the EMLA sham stays occluded for 30 min, which may unmask allocation to clinicians. Describe tactile/visual masking steps or harmonise application times.

i) Author’s Response

Thank you for the helpful comment. We confirm that the LEMAP’s maltose microneedle matrix dissolves within about 2 minutes. The patch will be nevertheless kept in place for 30 minutes to standardise application time between the LEMAP and control (sham patch) groups, preserve blinding (masking) between study arms, and allow sufficient time for dermal uptake of lignocaine.

ii) Manuscript changes (In Methodology, Trial Status Subsection)

“Although LEMAP fully dissolves within approximately 2 minutes, the procedurist will keep the patch in place for 30 minutes to standardise application time, maintain blinding, and ensure adequate dermal uptake before cannulation.” (Pages 21-22, Lines 384-387)

5) Reviewer’s Comment 5: Methodological detail for replication – Device fabrication steps are outlined, but key specifications (needle density, tip radius, residual lignocaine after dissolution) are missing. Likewise, SCAI sensor placement/calibration and VAS ruler calibration should be elaborated to enable replication.

i) Author’s Response

Thank you for your highly relevant and cogent suggestions. We have added the missing device specifications and clarified our dosing estimate for lignocaine. We now state that the microneedle tip density is 225 needles per cm², tip radius 2 μm and base radius 50 μm. We explain that residual lignocaine is typically about 20 percent of the initial load post-application, so about 80 percent is potentially delivered, and we show how this is approximated using the conical volume formula.

We have also added explicit procedures for SCAI sensor placement and calibration and for VAS ruler calibration to enable replication. These insertions are placed in the Intervention Characteristics and Trial Outcomes sections of the manuscript.

ii) Manuscript changes (In Several Sections)

a) Methodological Section (In Physical Descriptions of the Microneedle Array Patch Subsection)

“Each LEMAP contains 225 microneedles per cm². The tip radius is 2 μm and the base radius is 50 μm. Lignocaine load is assumed to be uniformly distributed within each microneedle. The total lignocaine volumes per microneedle before application and after application are approximated using the conical volume formulae, V= 1/3 πr^2 h and V= 1/3 πr^2 h^*, with r as the base radius, h as the microneedle height before LEMAP application and h^* after LEMAP application. The maltose matrix volume was considered as negligible for lignocaine dose calculation. Post-application, about 20 percent of the initial lignocaine typically remains undissolved within the microneedle, so approximately 80 percent is potentially delivered to the skin. This geometric approximation is used only to standardise reporting of dose delivery since pharmacokinetic sampling in our Phase I trial remains the determinant of systemic exposure.” (Pages 16-17, Lines 255-264)

b) Methodological Section (In Trial Outcomes Subsection, Under Primary Endpoint) - For VAS ruler calibration

“For VAS instrument calibration and standardisation, we will use a 0-100 mm slider VAS ruler of fixed length. Before first use and weekly thereafter, the research team will verify the physical scale length against a well-calibrated and standardized steel ruler and confirm the 0 mm and 100 mm markings on the VAS ruler align 0 mm and 100 mm markings on the reference ruler. VAS ruler that fails this check will be replaced. The participant will set the slider unaided, the trained outcome assessor (medical officer) will read the value from the reverse side.” (Page 22-23, Lines 405-411)

b) Methodological Section (In Trial Outcomes Subsection, Under Primary Endpoint) - For SCAI Sensor Placement and Calibration

“The skin conductance algesimeter index (SCAI), measured in microSiemens per second (μS/s), will be obtained using PainMonitor™ (Med-Storm Innovation AS, Oslo, Norway) (Figure 7). The device consists of a sensor will be attached to the hypothenar eminence of the hand of the limb not subjected to venepuncture. The skin will be cleaned with alcohol swab and allowed to air dry before PainMonitor™’s sensor is attached. The single-use sensor will be applied with firm, even contact according to the manufacturer’s instructions. The wrist and hand will be supported to minimise motion. The same limb and site will be used across participants whenever feasible.” (Page 23, Lines 412-419)

“For SCAI calibration, the operator will perform the PainMonitor™’s built-in quality check, confirm stable baseline conductance and acceptable signal quality and document ambient temperature and obvious sources of electrical or motion artefact. A baseline of at least 60 seconds will be recorded before venepuncture with the participant resting quietly. The post-venepuncture SCAI value obtained at one minute will be the primary time point, as prespecified above. If artefact is detected during this 60-second period, the operator will mark the trace and repeat a one-minute acquisition once the artefact resolves, without revealing treatment allocation. The same device model will be used throughout and operators will follow a standard operating procedure that includes periodic retraining and inter-operator checks.” (Page 23, Lines 424-432)

6) Reviewer’s Comment 6a: Primary comparisons use t/Mann-Whitney tests, then linear regression with covariate selection by multiple strategies. Mixing stepwise (AIC/BIC), purposeful selection and clinical judgement is overly flexible and can inflate false-positive rates. Specify a single, preferably pre-defined strategy—e.g., include only clinically justified pre-specified covariates plus any baseline imbalance >10 %.

i) Author’s Response

Thank you for your comment on this pertinent point about the inflated type I (false-positive) error rate from mixing variable selection strategies. We agree and have simplified the analysis plan to a single, pre-defined approach. The primary multiple linear regression (MLR) will adjust for a fixed set of clinically justified covariates (baseline VAS, baseline SCAI, age, sex, BMI). No stepwise/AIC/BIC/purposeful selection will be used. To address any meaningful baseline imbalance, we will add any other measured baseline covariate that shows a standardized difference of more than 10% between arms to the model under a pre-specified rule. We will report baseline balance using standardized differences and will not use significance testing of baseline characteristics to drive model specification. This aligns with the TRIPOD guideline to avoid data-driven selection and to pre-specify predictors to limit overfitting and enhance reproducibility [1].

ii) Manuscript changes (In Methodology, Statistical Analysis Plan Subsection)

“To control and adjust the effects of clinically-justified pre-specified covariates (confounders) such as baseline pain scores (baseline VAS and baseline SCAI), body mass index (BMI), gender, multiple linear regression (MLR) analysis will be used. We will not use automated or data-driven variable selection (no stepwise/AIC/BIC or purposeful selection). In addition, if any other measured baseline covariate shows a standardised difference >10% between arms, it will be added under this pre-specified rule to mitigate chance imbalance. Baseline balanc

---

## [Decision Letter · Decision Letter 1]

20 Oct 2025

The efficacy and safety of lignocaine-embedded dissolvable microneedle versus EMLA  for topical analgesia in adults undergoing venepuncture: A Single-Centre, Parallel-Group, Double-Blind Randomised Clinical Trial Protocol in a Tertiary Care Setting

PONE-D-25-38618R1

Dear Dr. Abdul Jalal,

We’re pleased to inform you that your manuscript has been judged scientifically suitable for publication and will be formally accepted for publication once it meets all outstanding technical requirements.

Kind regards,

Jeffrey Pradeep Raj, MBBS.,MD.,PG Dip Fly Med, PG Dip Biostats, MBA

Academic Editor

PLOS ONE

Additional Editor Comments (optional):

Reviewers' comments:

Reviewer's Responses to Questions

**Comments to the Author**

1. Does the manuscript provide a valid rationale for the proposed study, with clearly identified and justified research questions?

Reviewer #1: Yes

Reviewer #2: Yes

2. Is the protocol technically sound and planned in a manner that will lead to a meaningful outcome and allow testing the stated hypotheses?

Reviewer #1: Yes

Reviewer #2: Yes

3. Is the methodology feasible and described in sufficient detail to allow the work to be replicable?

Reviewer #1: Yes

Reviewer #2: Yes

4. Have the authors described where all data underlying the findings will be made available when the study is complete?

Reviewer #1: Yes

Reviewer #2: Yes

5. Is the manuscript presented in an intelligible fashion and written in standard English?

Reviewer #1: Yes

Reviewer #2: Yes

You may also provide optional suggestions and comments to authors that they might find helpful in planning their study.

Reviewer #1: I find that the authors have clearly addressed the previous comments and substantially improved the manuscript. I have no further remarks and wish the team the best of luck with the study.

Reviewer #2: The authors have comprehensively addressed all the comments and concerns raised in my previous review of the manuscript

**Do you want your identity to be public for this peer review?** For information about this choice, including consent withdrawal, please see our Privacy Policy

Reviewer #1: No

Reviewer #2: No

---

## [Editor Report · Acceptance letter]

PONE-D-25-38618R1

PLOS ONE

Dear Dr. Abdul Jalal,

I'm pleased to inform you that your manuscript has been deemed suitable for publication in PLOS ONE. Congratulations! Your manuscript is now being handed over to our production team.

Kind regards,

on behalf of

Dr. Jeffrey Pradeep Raj

Academic Editor

PLOS ONE